# Uncertainty as a criterion for SOTIF evaluation of deep learning models in autonomous driving systems

**Ho Suk**
School of Integrated Technology
Yonsei University
sukho93@yonsei.ac.kr

**Shiho Kim**
School of Integrated Technology
Yonsei University
shiho@yonsei.ac.kr

## Abstract

Ensuring the safety of deep learning models in autonomous driving systems is crucial. In compliance with the automotive safety standard ISO 21448, we propose uncertainty as a new complementary evaluation criterion to ensure the safety of the intended functionality (SOTIF) of deep learning-based systems. To evaluate and improve the trajectory prediction function of autonomous driving systems, we utilize epistemic uncertainty, quantified by a single forward pass model with consideration for constraints on resources and response time, as a criterion. Experimental results with data collected from the CARLA simulator demonstrate that uncertainty criterion can detect functional insuffiencies in unknown driving scenarios which potentially hazardous, and eventually induce extra learning.

## 1   Introduction

In the field of machine learning, the deep learning has made several technological breakthroughs over the past decade, rapidly improving its performance in a variety of tasks. Today, deep learning is widely applied to the implementation of perception and planning subsystems, realizing advances of high-level driving automation. Now, in addition to the performance of deep learning, the issue of uncertainty is also being seriously considered. The issue of uncertainty should also be discussed in the field of autonomous driving [10].

Safety is the most important factor in automobiles. Therefore, the automotive industry develops automobiles in compliance with automotive safety standards such as ISO 26262 and ISO 21448. ISO 26262 aims to ensure the functional safety of the system, and ISO 21448 aims to ensure the safety of the intended functionality (SOTIF) of the system. ISO 26262 only deals with faults at the software level of the deep learning model. ISO 21448, released to complement the insufficient coverage of ISO 26262, deals with insufficiencies in the intended functionality of the model. ISO 21448 emphasizes that pass/fail criteria for training and testing of deep learning should be well specified. However, it is very difficult to fully validate deep learning models that have non-linear characteristics and lack formal verification methods, so complementary validation is necessary.

Deep learning suffers from uncertainty, and ISO 21448 briefly mentions the uncertainty issue of machine learning. Nevertheless, the potentially hazardous uncertainty issue is not actively considered in the design of autonomous driving systems. In this paper, we propose the idea of quantifying uncertainty from the output of a deep learning model used in autonomous driving to utilize uncertainty as a pass/fail criterion of the deep learning model and as a fallback criterion for authority to control the driving. Section 2 explains standards for safety, types of uncertainty, and uncertainty quantification methods. Section 3 introduces a method to apply uncertainty quantification to trajectory prediction functions for autonomous driving and presents experimental results. Section 4 discusses future works and concludes the paper.

Workshop on Bayesian Decision-making and Uncertainty, 38th Conference on Neural Information Processing Systems (NeurIPS 2024).

## 2 Background and related works

### 2.1 Standards for automotive safety

**ISO 26262**  ISO 26262 is an international standard that guides the development of automotive systems by ensuring functional safety. According to the ISO 26262, safety goals are decomposed into functional safety requirements, and the system are developed to meet those specifications. To prevent hazards caused by faults from becoming unreasonable risks, safety measures that satisfy functional safety requirements must be implemented. By verifying and validating that the safety measures meet the requirements, the functional safety of the automotive system can be ensured.

If a deep learning model in ADAS and autonomous driving system is viewed as a type of software that receives inputs and generates outputs through a series of computing operations, compliance with ISO 26262 can ensure the functional safety of the model by checking that there are no problems with the software architecture or code. However, deep learning models completed by training deep neural networks have a completely different paradigm from existing automotive software. The prediction process and performance limitations of deep learning are not intuitive, so it is very difficult to explain them in specifications. It is impossible to make deep learning-based software ensure the safety of automobiles with the existing ISO 26262 standard alone.

**ISO 21448**  To make up for the coverage of ISO 26262, the ISO 21448 standard, which deals with SOTIF, was published. ISO 21448 presumes that the functional safety is guaranteed, and addresses risks that may occur in a fault-free system. Functional insufficiencies in a system may lead to hazards, which may cause risks. ISO 21448 aims to achieve SOTIF by reducing unreasonable risks as much as possible. In the SOTIF workflow, acceptance criteria are defined to prove the absence of unreasonable risks for identified hazards. SOTIF can be guaranteed by providing evidence that the acceptance criteria have been achieved with sufficient confidence through validation such as testing in known and unknown scenarios. See Figure 2 and 3 in Appendix A for details of SOTIF activities.

Specifying the criteria to ensure SOTIF of a deep learning model is different from the case of general automotive software. For example, in a deep learning model that performs object detection or classification, false positive and false negative rates can be used as pass/fail criteria during the training and test phases. At the vehicle level, accidents per unit of driving distance can be used as criteria to identify whether there are any functional insufficiencies in the intended functionality of the perception subsystem. However, to prove evidence for SOTIF, complementary criteria are needed to validate a deep learning model with low interpretability. In this case, uncertainty can be a candidate.

### 2.2 Types of uncertainty

A probability vector that the classifier model outputs along with the prediction is difficult to serve as a meaningful metric to measure the confidence in the output worked out from the input data that is out of distribution from the training data [2]. To evaluate the safety of deep learning model in any distribution, uncertainty of the output needs to be quantified.

**Aleatoric uncertainty**  Aleatoric uncertainty is a metric that quantifies the ambiguity of data resulting from sensor noise and randomness of the driving environment. Since noise and randomness are inherent in the data, aleatoric uncertainty cannot be relieved by increasing the amount of training data. Aleatoric uncertainty is not a major scope in this paper because it is irreducible.

Table 1: Summary of characteristics of state-of-the-art uncertainty quantification methods.

| | # of models | # of inferences | # of trainable $\theta$ | Computational burden |
|---|---|---|---|---|
| Bayesian model | Single | Single | High | High (Posterior distribution) |
| Monte Carlo dropout | Single | Multiple | Moderate | High (Monte Carlo sampling) |
| Deep ensembles | Multiple | Multiple | High | High (Bootstrap aggregating) |
| Deterministic single forward pass | Single | Single | Moderate | Moderate |

**Epistemic uncertainty** Epistemic uncertainty is a metric that quantifies the lack of knowledge required for a model to explain data. In other words, it represents a limitation that the distribution of training data cannot properly approximate the real world. By updating the model by providing previously unseen data as additional training data, epistemic uncertainty can be decreased.

### 2.3 Approaches to uncertainty quantification

State-of-the-art methods for uncertainty quantification can be classified into four categories: Bayesian model, Monte Carlo dropout, deep ensemble, and deterministic single-forward-pass method [3]. These approaches are implemented quite distinctly from each other and have different characteristics, so it is recommended to choose the best method that fits the domain and task to which applying. Table 1 summarizes the comparison of the characteristics of the four methods.

Some studies applying uncertainty quantification to the autonomous driving domain have been published. Most of them utilized the deep ensemble architecture [11, 4, 8, 9, 12]. However, compared to a typical deep neural network, there is a trade-off between the benefit and inefficiency of uncertainty quantification in that multiple networks forming the ensemble must be trained and inferred at once [6]. Since the autonomous driving system is executed on edge hardware, it is required to be executed with as few resources and computing power as possible. In addition, it is required to provide fast responses with the shortest possible run time for safety reasons. Considering efficiency, we judged that a deterministic single-forward-pass approach with a small number of parameters and low computational burden is suitable, so we selected Deep Deterministic Uncertainty (DDU) [7], one of the state-of-the-art methods, as a method for our uncertainty quantification.

## 3 Uncertainty as a criterion for SOTIF of deep learning-based trajectory prediction model

We applied uncertainty quantification to trajectory prediction, a key function required for decision making of autonomous vehicles. First, we collected driving data from the CARLA simulator as a dataset [1]. The input data of the model consists of the state vector of a vehicle and a bird's eye view image that expresses the surroundings of a vehicle. We only use a single model for training and inferencing. Our model is a ResNet-based structure. Residual block and spectral normalization improve generalization by well regularizing the feature space and maintaining sensitivity to input at an appropriate level. See Appendix B for details of our dataset and network architecture.

Our model predicts future velocity and future yaw of each vehicle. We classify the continuous values of velocity and yaw into small intervals, so the model predicts the one class with the highest probability. In addition to getting predictions, by feeding validation data to the frozen model, we can

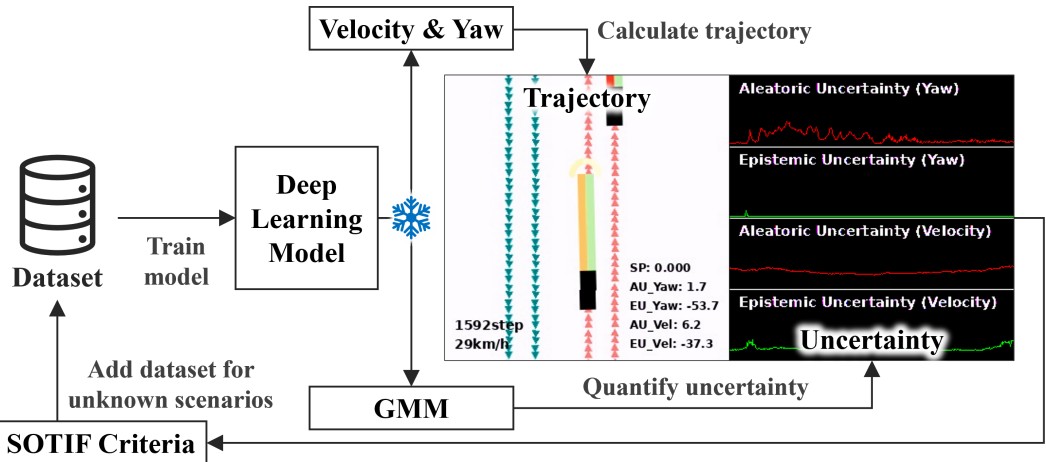

Figure 1: Simplified workflow of proposed idea that utilizes quantified uncertainty as a pass/fail criterion for SOTIF evaluation of a deep learning model for trajectory prediction task.

Table 2: Comparison of epistemic uncertainty mean in two different driving scenarios. We normalize the uncertainty to have a value greater than 0. The epistemic uncertainty mean is the average of the quantified uncertainty over the scenario execution time. The values are expressed relatively, with the value in the known scenario as 1.

| Scenario type | Urban (Known) | Highway (Unknown) | Highway (Extra Training) |
|---|---|---|---|
| Epistemic uncertainty mean | 1 | 3.42 | 1.51 |

obtain two Gaussian mixture models (GMM) representing the feature space for velocity and yaw, respectively. See Figure 4 in Appendix C for details of GMM that we obtain. The obtained GMM is used to quantify epistemic uncertainty of predicted velocity and yaw via Gaussian discriminant analysis (GDA). Epistemic uncertainty is the marginal likelihood of the feature space represented by the density model such as GMM. We interpolate the predicted velocities and yaws to compute the future trajectory. If the uncertainty of velocity is high, it can be interpreted that the longitudinal uncertainty of the trajectory is high, and if the uncertainty of yaw is high, it can be interpreted that the lateral uncertainty of the trajectory is high. Figure 1 describes a workflow that quantifies uncertainty in a trajectory prediction task and uses it as SOTIF criteria.

As described in Section 2, the evaluation of deep learning models should be performed in both known and unknown scenarios. In the case of known scenarios, metrics commonly used in the evaluation of deep learning models can be defined as target requirements. On the other hand, in the case of unknown scenarios, it is difficult to set criteria in advance because there are many long tail cases in the real world. For example, it is impossible to create criteria for all driving situations that the ego vehicle may encounter. ISO 21448 simply recommends a large amount of testing to reduce unknown scenarios, and suggests cumulative test duration and scenario types as criteria.

We propose that epistemic uncertainty can be used as a criterion to identify unknown scenarios. When applied to our deep learning model, if the epistemic uncertainty of the predicted trajectory provided by the model is estimated to be high in a specific scenario during the evaluation process, it can be interpreted that the autonomous driving function is likely to fail in that scenario due to incorrect predictions caused by insufficient model knowledge. As a pass/fail criterion of the model, epistemic uncertainty can play a role of indicator for necessity of additional training in the unknown scenarios. In our experiment, highway environment corresponds to an scenario where our model is out of distribution for our model. Table 2 shows that uncertainty is measured high in highway and can be judged as unknown scenarios. It also show that uncertainty can be lowered after extra training. Please see Figure 5 and 6 in Appendix D for an example of our results.

If the model is judged to satisfy the pass/fail criteria and ensure SOTIF, the model is deployed as an autonomous driving system and operated on real world. Autonomous vehicles may encounter long tail cases that violate SOTIF. Since the deep learning model is designed to produce output even in situations where there is uncertainty due to insufficient knowledge, it may make incorrect predictions. In this case, epistemic uncertainty can serve as a start signal for the fallback mechanism. When the estimated uncertainty is high so functional insufficiency or potential hazard is expected, to prevent an accident, the system can hand over the control to the human driver or take emergency procedure. See Figure 3 in Appendix A for details of uncertainty as criteria in SOTIF activities.

## 4 Conclusion

In the trajectory prediction task, we quantify uncertainty via a deep learning model with a single forward pass. Then, we use the uncertainty as a criterion for SOTIF evaluation. If the epistemic uncertainty exceeds the criterion, we consider that the scenario could be potentially dangerous, so we collect additional data and perform extra training. The experimental results show that uncertainty has the possibility to be utilized as a criterion to judge the functional insufficiencies of autonomous driving systems in unknown scenarios.

There is also a study that performed online learning of an end-to-end autonomous driving system using uncertainty [5]. Uncertainty quantification appears to have the potential to be used in the evaluation and learning of autonomous driving systems. However, further development is still needed to address the resource and time consumption issues caused by uncertainty quantification process.

## Acknowledgments and Disclosure of Funding

This work was supported by Institute of Information & communications Technology Planning & Evaluation (IITP) grant funded by the Korea government(MSIT) (No.RS-2021-II211352, Development of technology for validating the autonomous driving services in perspective of laws and regulations).

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

# A  SOTIF activities described in ISO 21448

The goal of SOTIF activities is to maximize the number of known scenarios and minimize the number of hazardous scenarios. As a result, the confidence of safety will be increased by increasing known scenario set, and the residual risk will be decreased by decreasing hazardous scenario set. These activities ultimately ensure SOTIF. Figure 2 describes the evolution of scenarios by the progress of SOTIF activities.

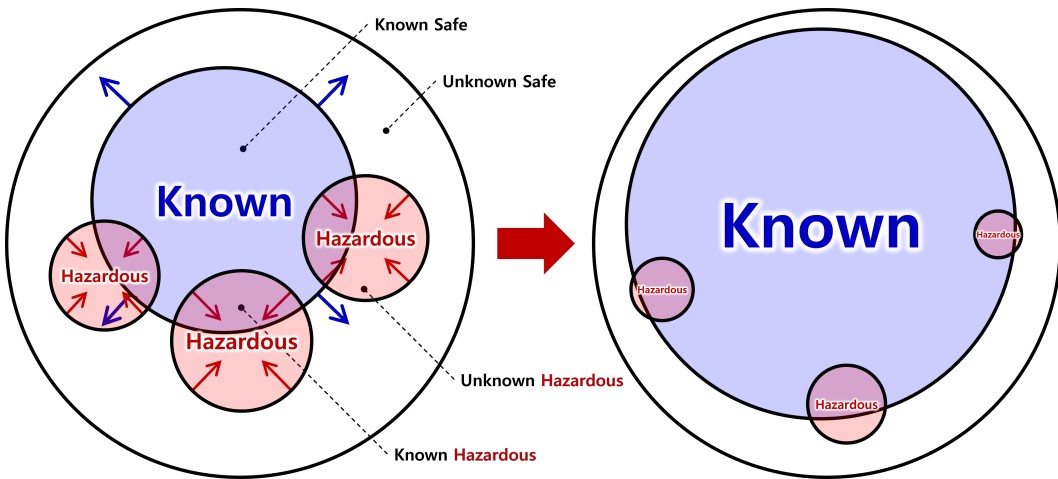

Figure 2: Evolution of scenarios resulting from SOTIF activities. Circles represent distributions of set of scenarios. Arrows around the circles indicate increases or decreases of scenario set due to SOTIF activities. Scenarios are divided into 4 categories: known safe, unknown safe, known hazardous, and unknown hazardous.

We argue that uncertainty can be utilized in 2 types of phases that constitute SOTIF activities in ISO 21448. In the evaluation of unknown scenarios, uncertainty can be utilized as a pass/fail criterion for a deep learning model. And in the operation phase activities, uncertainty can be utilized as a fallback criterion for an autonomous driving system based on a deep learning model. Figure 3 shows the applicability of uncertainty to SOTIF activities.

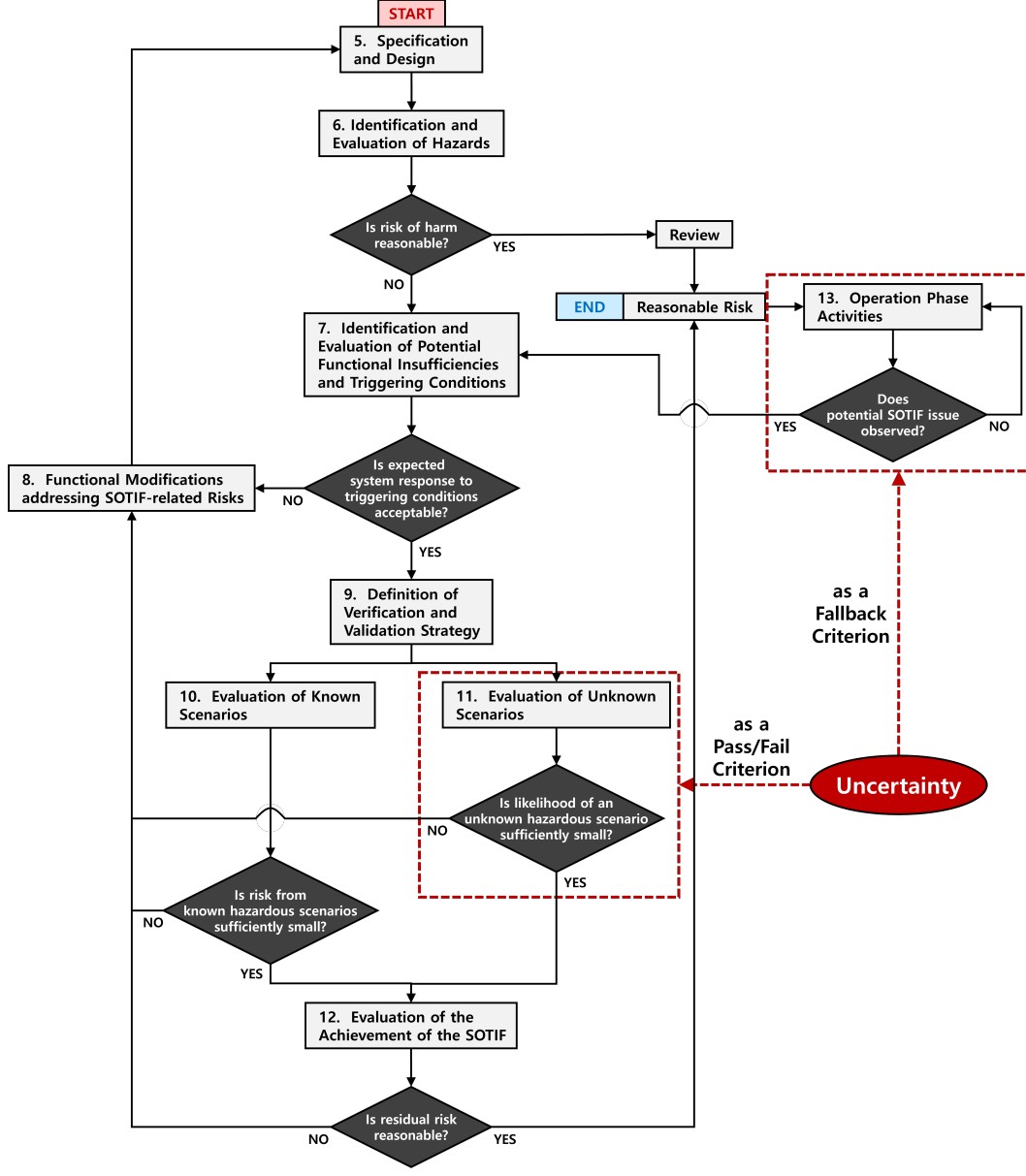

Figure 3: Applicability of uncertainty to SOTIF activities as pass/fail criterion and fallback criterion for deep learning model-based autonomous driving system. It is represented by a box drawn with a red dotted line.

## B  Details of dataset and network architecture

Our state vector data of vehicles were collected at 20Hz on the CARLA simulator. Table 3 shows data components of the state vector. The environment where training data was collected is Town03 map, an urban environment consisting of roads with up to 4 lanes, several intersections, and roundabouts, The environment where data for evaluating the model in unknown scenarios was collected is Town06 map, a highway environment consisting of roads with up to 12 lanes.

The bird's eye view image is not collected directly from the CARLA simulator but is drawn abstractly using the collected road and vehicle data. The bird's eye view image shows the road structure by indicating the virtual center line of each lane, and also shows surrounding vehicles within a certain range from a vehicle.

Table 3: Data components constituting state vector of vehicles collected from the CARLA simulator.

| Attribute | Data Unit |
| --- | --- |
| Position x | meter |
| Position y | meter |
| Yaw | degree |
| Velocity x | m/s |
| Velocity y | m/s |
| Angular velocity | rad/s |
| Acceleration x | $m/s^2$ |
| Acceleration y | $m/s^2$ |
| Is at a stop line | true or false |
| Traffic light status | red or yellow or green |

Data from driving environments that are not included in training dataset are used to evaluate whether epistemic uncertainty increase in unknown scenarios. Although this study used preprocessed simulation data as input, our prediction model has the potential to be integrated with a perception model that processes raw data from sensor to construct an autonomous driving system.

Our ResNet-based model has a shallow depth consisting of 4 convolution blocks followed by 4 fully connected blocks. Each block has 2 layers, and a residual connection exists between blocks. Spectral normalization is applied to each layer. The very first layer of the convolution block gets a bird's eye view image as an input. The very first layer of the fully connected block gets a vector, that concatenates the flattened convolution feature map and the state vector, as an input.

# C Gaussian mixture model representing feature space of prediction model

We calculated class-wise mean and covariance using feature embedding, which is the input to the output layer, the last layer of our network, and then fit the Gaussian mixture model (GMM) to these classes. The number of Gaussian distributions that make up the GMM corresponds to the number of classes. Figure 4 shows 2D and 3D visualizations of the GMM, which represents the feature space of our prediction model through principal components analysis (PCA).

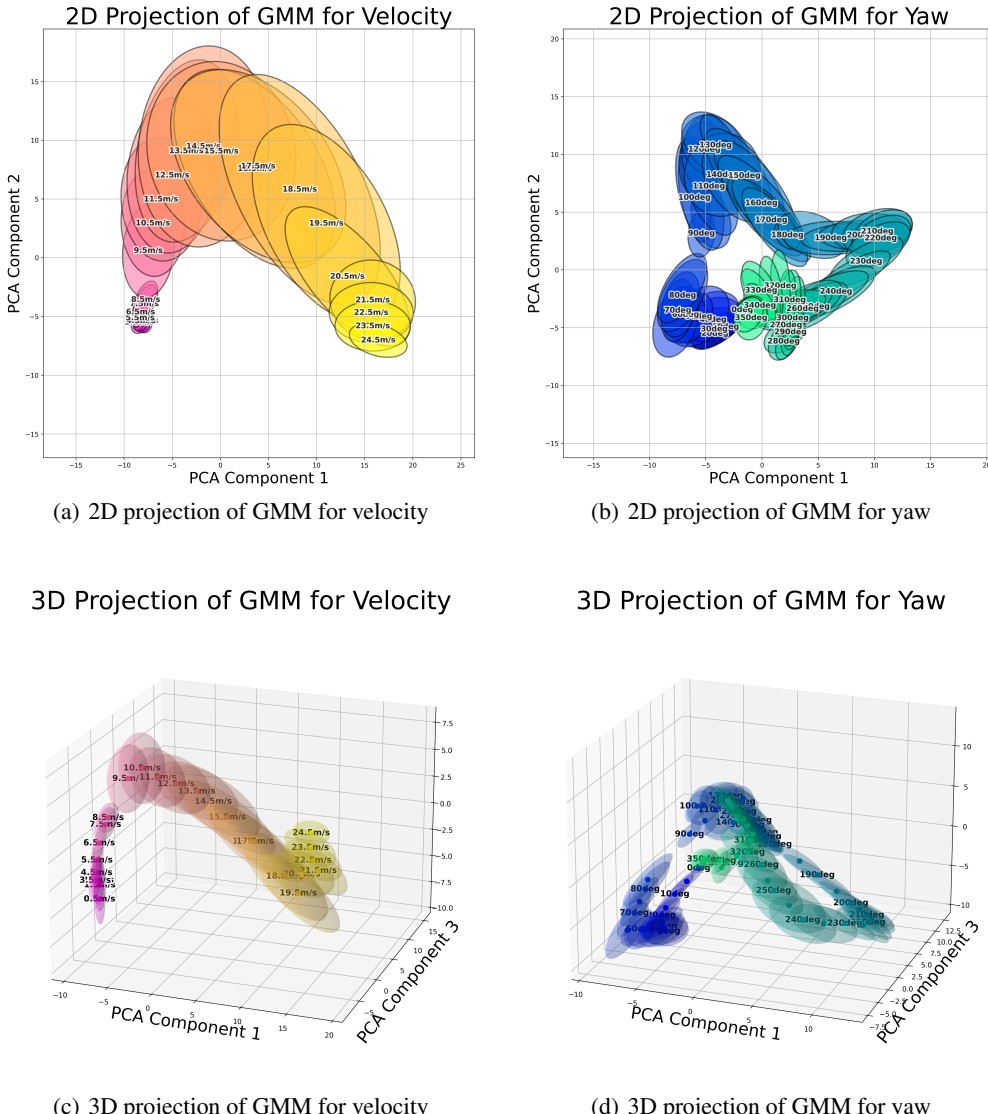

(a) 2D projection of GMM for velocity

(b) 2D projection of GMM for yaw

(c) 3D projection of GMM for velocity

(d) 3D projection of GMM for yaw

Figure 4: 2D and 3D visualization of GMM representing the feature space of our model.

## D   Example describing predicted trajectory and quantified uncertainty in CARLA simulation

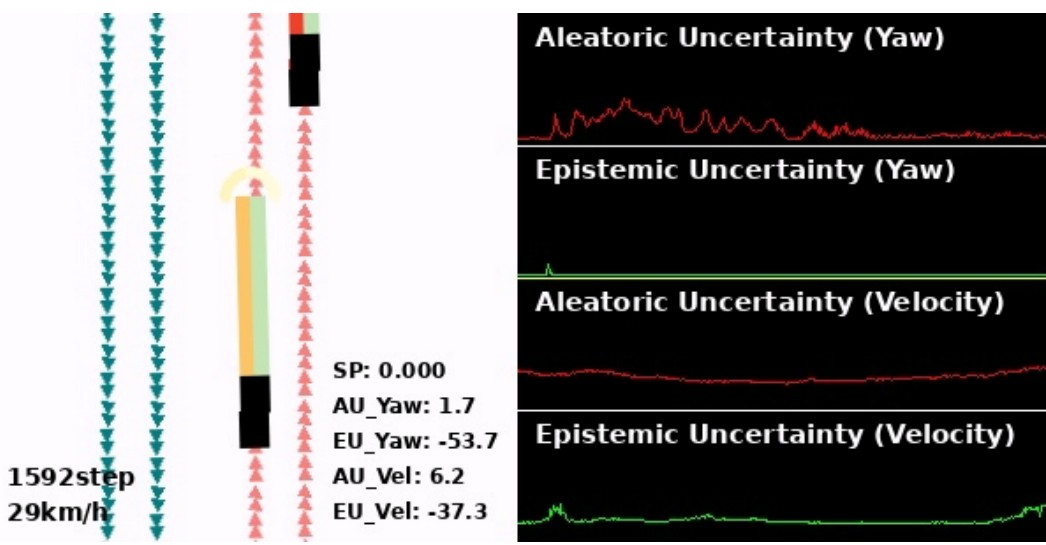

Figure 5: Predicted trajectory and quantified uncertainty in Town03 map (urban environment) of CARLA simulator. Since it is an environment where the model has been trained, uncertainty is low and trajectories are predicted normally.

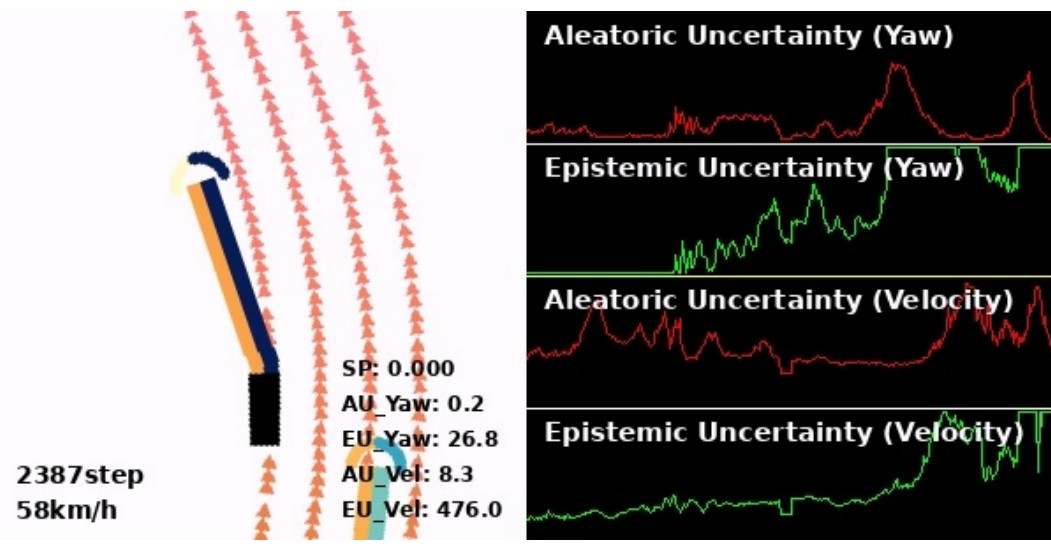

Figure 6: Predicted trajectory and quantified uncertainty in Town06 map (highway environment) of CARLA simulator. Since it is an environment where the model has not been trained, uncertainty is high and trajectories are predicted incorrectly, different from the actual situation. From the SOTIF perspective, it corresponds to an unknown hazardous scenario.

