# OpenReview forum: "Uncertainty as a criterion for SOTIF evaluation of deep learning models in autonomous driving systems"
_NeurIPS.cc/2024/Workshop/BDU — NeurIPS BDU Workshop 2024 Poster_

### Official Review · Reviewer_1fQY · 2024-09-22
**Uncertainty as a Criterion for SOTIF Evaluation of Deep Learning Models in Autonomous Driving Systems**

**Rating:** 6
**Confidence:** 3

**Review:**

1. Summary and contributions:

The main contribution of this paper is proposing uncertainty as an evaluation criterion of deep learning models in autonomous driving areas. The authors clarify the scope of aleatoric uncertainty and epistemic uncertainty and explain the reason of choosing Deep Deterministic Uncertainty as the method for uncertainty quantification.

2. Strengths:

The paper is generally well-written and structured clearly. The authors clearly present the motivation of the paper and overall workflows of the experiments which makes me quite easy to read. Besides,  experimental results demonstrate that this proposed uncertainty criterion effectively identifies potential functional deficiencies.


3. Weaknesses:

The experiments are somewhat insufficient that do not totally enthuse me. To further examine the effectiveness of the uncertainty metric, I think that including comparative experiments assessing uncertainty across different trajectory prediction algorithms can further prove the effectiveness of the proposed metric and also provide readers with a broader perspective on how uncertainty manifests across various algorithms.

---

### Official Review · Reviewer_i43t · 2024-10-08
**This work explores the use of uncertainty as a criterion for ensuring the Safety of the Intended Functionality (SOTIF) in deep learning-based systems for autonomous driving**

**Rating:** 8
**Confidence:** 3

**Review:**

This work explores the use of uncertainty as a criterion for ensuring the Safety of the Intended Functionality (SOTIF) in deep learning-based systems for autonomous driving. By leveraging epistemic uncertainty, the authors propose a pass/fail evaluation method for trajectory prediction tasks, aiming to improve safety and reliability in both known and unknown driving scenarios.

Pros
- The integration of deep learning in autonomous vehicles is an ongoing challenge, and focusing on SOTIF complements the widely accepted ISO 26262 standard by addressing functional insufficiencies.
- Novel use of uncertainty: The authors apply uncertainty quantification (specifically epistemic uncertainty) to detect potentially hazardous driving scenarios, which could enable real-time safety assessments.
- The application of uncertainty quantification as a fallback mechanism is innovative and well-justified, providing a robust method to identify potential hazards in unknown driving scenarios.
- The paper aligns well with the focus on theoretical contributions and the workshop's theme, offering a strong framework for uncertainty quantification in deep learning models, particularly suited to safety-critical systems.
- The experimental results, based on the CARLA simulator, show that their uncertainty criterion can indeed detect functional insufficiencies and suggest corrective measures like additional training, which is validated by reduced uncertainty in subsequent tests.

Cons
- The authors could talk more and perform a comparative analysis of other methods over DDU

---

### Official Review · Reviewer_B4sw · 2024-10-09
**Review of Submission 114**

**Rating:** 4
**Confidence:** 3

**Review:**

## Summary:

This paper proposes using epistemic uncertainty as a criterion for evaluating the Safety of the Intended Functionality (SOTIF) of deep learning models in autonomous driving systems, in alignment with the ISO 21448 standard. The authors apply Deep Deterministic Uncertainty (DDU) to quantify uncertainty in a trajectory prediction task using data from the CARLA simulator. They suggest that high epistemic uncertainty can identify functional insufficiencies in unknown driving scenarios, which can be mitigated through additional training.

## Strengths:
- The paper addresses a critical aspect of autonomous vehicle safety by proposing a method that aligns with ISO 21448, which is significant for industry practices.
- The authors consider a realistic setting while building the problem statement, e.g., assuming limited computational resource during deployment.

## Weaknesses:
- While the paper references ISO 21448, it lacks a detailed integration of the standard into the methodology and does not demonstrate practical implementation within the SOTIF framework.
- The paper provides insufficient details about the model architecture, training procedures, hyperparameters, and the implementation of uncertainty quantification.
- There are no quantitative metrics presented to substantiate the claims. The paper lacks evaluation of prediction accuracy, uncertainty calibration, or the impact on safety performance. Further, there is no comparison with other uncertainty quantification methods or baseline models, making it difficult to assess the effectiveness of the proposed approach.

---

### Decision · Program_Chairs · 2024-10-09

Accept (Poster)